# Evaluation, Description and Magnitude of Readmission Phenomenon in Azienda Ospedaliero Universitaria Pisana (AOUP) for Chronic-Degenerative Diseases in the Period 2018–2021

**DOI:** 10.3390/healthcare11050651

**Published:** 2023-02-23

**Authors:** Matteo Filippi, Erika Del Prete, Ferruccio Aquilini, Michele Totaro, Francesca Di Serafino, Sara Civitelli, Giulia Geminale, David Rocchi, Nunzio Zotti, Angelo Baggiani

**Affiliations:** 1The Azienda Ospedaliero Universitaria Pisana (AOUP), 56100 Pisa, Italy; 2Department of Translational Research and the New Technologies in Medicine and Surgery, University of Pisa, 56123 Pisa, Italy

**Keywords:** readmissions, health economics, Diagnosis Related Group, chronic diseases, continuity of care

## Abstract

Background: Readmissions are hospitalizations following a previous hospitalization (called index hospitalization) of the same patient that occurred in the same facility or nursing home. They may be a consequence of the progression of the natural history of a disease, but they may also reveal a previous suboptimal stay, or ineffective management of the underlying clinical condition. Preventing avoidable readmissions has the potential to improve both a patient’s quality of life, by avoiding exposure to the risks of re-hospitalization, and the financial well-being of health care systems. Methods: We investigated the magnitude of 30 day repeat hospitalizations for the same Major Diagnostic Category (MDC) in the Azienda Ospedaliero Universitaria Pisana (AOUP) over the period from 2018 to 2021. Records were divided into only admissions, index admissions and repeated admission. The length of the stay of all groups was compared using analysis of variance and subsequent multi-comparison tests. Results: Results showed a reduction in readmissions over the period examined (from 5.36% in 2018 to 4.46% in 2021), likely due to reduced access to care during the COVID-19 pandemic. We also observed that readmissions predominantly affect the male sex, older age groups, and patients with medical Diagnosis Related Groups (DRGs). The length of stay of readmissions was longer than that of index hospitalization (difference of 1.57 days, 95% CI 1.36–1.78 days, *p* < 0.001). The length of stay of index hospitalization is longer than that of single hospitalization (difference of 0.62 days, 95% CI 0.52–0.72 days, *p* < 0.001). Conclusions: A patient who goes for readmission thus has an overall hospitalization duration of almost two and a half times the length of the stay of a patient with single hospitalization, considering both index hospitalization and readmission. This represents a heavy use of hospital resources, about 10,200 more inpatient days than single hospitalizations, corresponding to a 30-bed ward working with an occupancy rate of 95%. Knowledge of readmissions is an important piece of information in health planning and a useful tool for monitoring the quality of models of patient care.

## 1. Introduction

In the last decade, facing an increasing complexity of care and the development of healthcare systems, we have witnessed a “patient readmission” phenomenon. With these words, we refer to a new admission of a patient discharged before from another hospital or healthcare setting [1]. Time cut-off and inclusion/exclusion standards are different among countries. In Italy, for example, we can talk of readmission when the same patient stays in a ward more than 1 day, after a previous admission (called “index hospitalization”) in the same healthcare setting. Readmissions are acceptable in some cases, such as frequent admission for chemotherapy treatments, but in other cases they could be a warning of some clinical issues. A premature discharge could not permit adequate patient management, increasing infectious exposure and death risk. Simultaneously, hospital spending and resources are compromised, as shown by the Management and Health (Management e Sanità, MeS) Laboratory of Sant’Anna School of Advanced Studies [2].

In the United States of America (USA), readmission has been a public health concern since the early 1980s, mainly for older adults related to their engagement in the Medicare program [3]. A few studies examined whether some interventions could prevent the readmission of those patients with chronic conditions such as diabetes [4,5] or chronic obstructive pulmonary disease (COPD) [6]. The main interest of readmission analysis was its use as a quality index, as well as the evaluation of rates and costs. In the late 1980s, the Prospective Payment System (PPS) began to be a concern for a likely increase in premature discharges, since health structures are paid by the Diagnosis Related Group (DRG) [7]. The same concern arose when the DRG system spread in Europe [8]. However, the impact of PPS on readmission rates was not supported by empirical evidence [9]. With the development of predictive models in public health, we can assume that readmissions are preventable by 10–50% of the total amount [10] and we can evaluate even disease-specific readmission rates [11]: heart failure, ischemic cardiac disease, myocardial infarction, COPD and pneumonia are the main focus for health management. The Centers for Medicare and Medicaid Services (CMS) in 2009 showed that pneumonia, heart failure and myocardial infarction were 20% of total admission in healthcare settings. In 2012, the US Government approved the Hospital Readmission Reduction Program (HRRP), which aimed to empower patients and caregivers in reducing readmission. The payment of healthcare structures is now based on the quality of care, rewarding hospitals with a lower 30-day readmission rate [12].

In Italy, the National Health Service (Servizio Sanitario Nazionale, SSN) is a separate case when we consider the Prospective Payment System (PPS) effects on health services [13]. DRGs were introduced in Italy in 1994 and the Tuscany Region has collaborated with the MeS Laboratory of the Sant’Anna School of Advanced Studies in the “System of performance evaluation” since 2006 [14,15,16,17,18]. This system was adopted by the other 12 Italian regions, founding the “Network of Regions” in 2008, where 27 University Hospitals are included. In this way, an annual report is shared with each region’s results, based on common indexes. Since 2011, this report has been available to all the stakeholders and the general public.

Furthermore, since 1991 in Italy there is a wide use of the Hospital Discharge Register (Scheda di Dimissione Ospedaliera, SDO), based on UHDDS (Uniform Hospital Discharge Data Set), which is a tool for collecting discharge information from healthcare settings [19]. In 2000, SDO was updated with ICD-9-CM classification [20]; it contains patients’ personal data, hospitalization features and clinical information. The importance of SDO concerns a uniform classification of hospital discharges, feeding a huge information flow that allows economic, management, clinic and epidemiologic evaluations.

Furthermore, by analyzing SDO information flow, we can evaluate different DRG-specific indicators, such as performance indicators, appropriateness indicators and control indicators. Among appropriateness indicators, we can find “Patient readmission in 30 days by same Major Diagnostic Category (MDC)”, which evaluates premature readmission in 30 days since the previous admission for the same (or similar) diagnosis. This indicator evaluates patient safety and the efficacy of resource employment.

In the USA, after local ventures [21], the national dedication to performance evaluation began with a pilot program of the “Joint Commission on Accreditation of Healthcare Organization” (JCAHO, later named “The Joint Commission”, JCI) [22]. In 2001, JCI, Centers for Medicare and Medicaid Services (CMS), American Hospital Association and others founded the Hospital Quality Alliance (HQA) as a means to sending performance data to CMS [23,24], with voluntary membership. In 2012, after the 2003 Medicare Modernization Act, quality indicators were integrated in CMS payment rates and published on the “Hospital Compare” website.

In Italy, quality indicators are regulated by Legislative Decree 502/1992, for accreditation purposes. Afterwards, the D.Lgs. 56/2000 introduced monitoring processes in health assistance, as a quality assurance system. Moreover, in 2001, the indicators and parameters to monitor the adherence of assistance essential levels by the regions were clarified. Among these parameters, we can find short and long stay rates, dismissal rates, hospitalization rates by specific surgical procedures, and others. The importance of quality evaluation in the SSN is now strengthened and in 2019 the Ministry of Health published the “New Guarantee System for monitoring health assistance” (Nuovo Sistema di Garanzia per il monitoraggio dell’assistenza sanitaria, NSG).

Concerning the heavy costs of patient readmission, about USD 15–20 billion [25] is spent yearly in the USA. In 2003, about 20% of Medicare patients were readmitted within 30 days. Readmission of HRPP target pathologies (myocardial infarction, cardiac failure and pneumonia) went from 21.5% in 2007 to 17.8% in 2015. Other pathologies went from 15.3% to 13.1% [26].

In the United Kingdom (UK), readmission rates in emergencies went from 8% in 1998 to 10% in 2006, being stable from 2006 to 2012 and decreasing in the next period [27]. However, there is variability among clinical areas, witnessing an increasing readmission rate for pneumonia, diabetes, cholecystectomy and hysterectomy. National Health System (NHS) data show a small increase in the national readmission rate, from 12.5% in 2013/2014 to 13.8% in 2017/2018 [28]. Moreover, readmission rates reached 14.9% in disadvantaged areas, but 12.7% in more advantaged areas.

Considering the readmissions, we can distinguish avoidable from unavoidable ones and our efforts concern the first sort [29]. A systematic review of 34 studies [30] shows that avoidable readmissions are 27.1%, even if they range from 5% to 79%. An observational study, conducted on 1000 patients of General Medicine readmitted in 30 days, ascertained that 27% are potentially preventable [31,32].

Many studies proposed clinical and demographic parameters to evaluate the risk of readmission and “high-risk patients” [33], as shown in Table 1.

With all that said, many efforts are spent on the research of screening tools to identify high-risk patients, but the results are not satisfying and predictive models have a low discriminating function. The literature asserts that it is not possible to predict readmissions and even most recent reviews show that predictive models, created for specific populations and conditions, are not standardizable [34].

For example, a screening tool is LACE index, which identifies patients with a readmission risk, but not avoidable readmissions [34,35]. This tool has been improved, creating LACE+, which contains more clinical parameters. Besides LACE/LACE+, another tool is HOSPITAL Score, specifically modeled to identify avoidable readmissions in 30 days with a computer algorithm [35] and it has a quite high discriminant capacity. Moreover, 8Ps is a tool created by the Society of Hospital Medicine to identify and stratify each risk and pair it with a risk-specific intervention.

Many factors increase readmission risk, and many factor are potentially avoidable or improvable [36], such as premature discharges, inadequate post-discharge support, delayed or missing follow-up, therapeutical mistakes, adverse drug reactions, and inefficient continuity of care.

Given this, as shown in a systematic review of 2011 [37], many efforts aim to reduce readmissions, improving predischarge and postdischarge interventions. In predischarge interventions, we can find patient education, medication reconciliation, discharge planning and scheduling of adequate follow-up. In postdischarge interventions are included all the follow-up measures such as telephone calls, home visits and ambulatory visits.

Other studies [38,39] underline how the combination of interventions can improve clinical outcomes and reduce readmission rates. Most efficient interventions are complex ones, aimed to give the patient a self-care capability.

The main goal of this paper is to evaluate readmission rates within 30 days, for the same MDC in a 4-year period—including data from the COVID-19 pandemic period—in Azienda Ospedaliero Universitaria Pisana (AOUP). The other purpose is the evaluation of the length of stay of readmitted patients to quantify the hospital burden. After an overall analysis, we focused on three clinical conditions: COPD, arrhythmia, and hypertension.

## 2. Materials and Methods

The study was performed in the Azienda Ospedaliero Universitaria Pisana (AOUP) (i.e., Pisa University Hospital), which represents the tertiary referral center for the “Toscana Nord-Ovest” local health unit. The administrative Hospital Discharge Records database of the AOUP represents the study data source. All records were coded with a medical DRG. We examined 30-day readmission rates with the same MDC diagnosis in AOUP, between 1 December 2017 and 31 January 2022.

The informed consent was applied for patients and healthcare personnel.

Variables included in the analysis were gender, age at the time of discharge, type of DRG (medical or surgical), length of stay, primary and secondary diagnosis codes according to the ICD-9-CM (International Classification of Diseases—ninth revision—Clinical Modification).

Descriptive statistics were used to describe the basic features of the data in the study. The variables used in the model were chosen according to availability in the HDR database.

The records were divided into three groups: only admissions (OA), index admissions (IA) to indicate the first admission of patients that were readmitted for the same MDC within 30 days after discharge, and repeated admissions (RA) to indicate admissions that followed the first one within 30 days from the previous discharge.

The length of stay of the three groups was compared using analysis of variance and subsequent multi-comparison tests. Outliers above the 95th percentile (corresponding to 20 days) were excluded from the analysis regarding the length of stay. The paired *t*-test was used to test for differences in the length of stay between IA and RA. The *t*-test for independent samples was used to test for differences in the length of stay between OA and RA.

One-way analysis of variance was used to evaluate differences between age groups in the mean length of stay of IA and OA. The same test has been performed to analyze differences in the readmission phenomenon between the pre-COVID-19 pandemic period (years 2018–2019) in comparison to the COVID-19 pandemic phase (years 2020–2021).

Association between categorical variables was assessed through the Chi-square test;, differences between continuous variables were compared using *t*-test and analysis of variance.

Multivariate logistic regression analysis was performed evaluating which variables influence readmissions. Dependent variables were obtained for IA versus OA groups.

We considered some binomial variables such as sex (male or female), type of DRG (medical, surgical), year groups (2018–2019, 2020–2021), and a multinomial variable (age groups: <36 years; 37–54 years; 55–67 years; 68–77 years; >77 years).

Three pathology groups were included in the analysis: chronic obstructive pulmonary disease (COPD), cardiac arrhythmias, hypertension.

Afterwards, the multicollinearity was evaluated in order to establish if variables are near perfect linear combinations of one another. We used the Variance Inflation Factor (VIF) command after the regression to check for multicollinearity. Variables whose VIF values are greater than 10 may merit further investigation. Tolerance, defined as 1/VIF, is used to check the collinearity degree.

A *p*-value ≤ 0.05 is regarded as evidence of a statistically significant result.

The software STATA 15.1 (StataCorp 4905 Lakeway Drive College Station, Texas 77845 USA) and MS Excel were used for all statistical analyses.

## 3. Results

Table 2 shows the general features of admissions in AOUP between 2018 and 2021.

During the study period, there were 167.792 hospital admissions, 8346 of which were repeated (4.97%). Outliers cover 5.6% (9345 out of 167.792) of the whole sample. It includes both male and female with different ages and pathologies. Splitting the study period in years, we could see that from 2018 to 2019 there was a slight reduction in readmission rates (from 5.36% in 2018 to 5.33% in 2019), whereas in 2020 and 2021 there was a more consistent reduction (4.57% in 2020 and 4.46% in 2021) (Figure 1).

A statistically significant reduction in readmission was observed in the COVID-19 pandemic period (years 2020–2021) in comparison to the pre-COVID-19 pandemic years (2018–2019) (*p* = 0.001).

From an overall evaluation of the four years, we could see that females were prevalent in only admissions (53.28% females versus 46.72% males); males were prevalent in index admissions and repeated admissions (45.46% females versus 54.54% males in RA and 46.47% females versus 53.53% males in IA).

The average age of patients with OA was 55.9 years (SD = ±24.3 years), while the average age of patients with IA was significantly higher, 61.7 years (SD = ±20.9 years). Dividing into quintiles by age group, we could see that OA had a similar subdivision in age groups; IA and RA were disproportionately older ages.

Analyzing the distribution in medical DRG and surgical DRG, OA had a majority of surgical DRG (55.60% surgical DRG versus 44.40% medical DRG); in contrast, IA and RA had a majority of medical DRG, especially in IA (74.57% medical DRG versus 25.43% surgical DRG).

We selected three clinical conditions according to the greater frequency of DRG in IA: COPD (DRG 4912x, 492xx, 4932x), cardiac arrhythmias (DRG 427), and hypertension (DRG 401). This choice was based on the fact that among IA the most frequent diagnoses were DRG 49121 chronic obstructive bronchitis, with acute exacerbation; DRG 42731 atrial fibrillation; DRG 4011 hypertension.

These diagnoses belong to the previously mentioned three clinical conditions.

To evaluate the length of hospital stays, first, we analyzed the distribution of the length of stays to eliminate admissions that were above the 95th percentile (corresponding to 20 days), because these outliers determined distortions of the remaining sample. From a global evaluation, we found that IA means a hospital stay was longer than an OA mean hospital stay (5.3 ± 4.41 days versus 4.68 ± 4.01 days). We conducted a longitudinal comparison between IA and RA of the same patients through a paired *t*-test. Excluding the outliers above 20 days, mean RA hospitalization (6.82 ± 8.22 days) was longer than mean IA hospitalization (5.25 ± 4.40 days). The difference, statistically significant, was 1.57 days (95% CI 1,36–1.78; *p* < 0.001).

Using a *t*-test for independent samples, we compared the lengths of the IA and OA stays. The mean IA length of stay (5.30 ± 4.41 days) was longer than the mean OA length of stay (4.68 ± 4.01 days). The difference, statistically significant, was 0.62 days (95% CI 0.52–0.72; *p* < 0.001). Moreover, IA hospitalization was longer than OA hospitalization both for males (5.36 days versus 4.98 days) and females (5.23 days versus 4.42 days).

To better understand the length of stay between patients that later underwent readmission (IA) and patients that have an only admission (OA), we analyzed the age breakdown. Even in this case, we excluded the outliers above 20 days. The results are presented in Figure 2. We can observe that the mean length of stay is longer for IA than OA for each age group.

Statistically significant differences between age groups in the mean length of stay of IA and OA were observed (*p* = 0.008).

After analyzing the overall sample, we decided to focus on three clinical conditions that were more frequent among index admissions: COPD (DRG 4912x, 492xx, 4932x), cardiac arrhythmias (DRG 427), and hypertension (DRG 401). We analyzed the mean length of stay and the age breakdown for each condition.

### 3.1. Chronic Obstructive Pulmonary Disease

We conducted a longitudinal comparison between IA and RA of the same patients with DRG 4912x, 492xx, 4932x, through a paired *t*-test. Excluding the outliers above 20 days, mean RA hospitalization (8.5 ± 8.70 days) was longer than mean IA hospitalization (6.88 ± 4.54 days). The difference, statistically significant, was 1.62 days (95% CI 0.64–2.60; *p* < 0.001).

Using a *t*-test for independent samples, we compared the length of stay of IA and OA for admission with DRG 4912x, 492xx, 4932x. The mean IA length of stay (7.01 ± 4.24 days) was longer than the mean OA length of stay (6.72 ± 4.33 days). Nevertheless, the difference was not statistically significant (*p* = 0.1). Even if the difference was not statistically significant, IA hospitalization was longer than OA hospitalization both for males (6.85 days versus 6.58 days) and females (7.29 days versus 6.98 days).

Considering the length of stay, we analyzed the age breakdown for patients with DRG 4912x, 492xx, 4932x. The results are presented in Figure 3. We can observe that the mean length of stay is longer for IA than OA for age groups above 68 years, but not for the 55–67 years group.

### 3.2. Arrhythmias

We conducted a longitudinal comparison between IA and RA of the same patients with DRG 427, through a paired *t*-test. Excluding the outliers above 20 days, mean RA hospitalization (7.89 ± 7.17 days) was longer than mean IA hospitalization (6.49 ± 4.21 days). The difference, statistically significant, was 1.39 days (95% CI 0.77–2.01; *p* < 0.001).

Using a *t*-test for independent samples, we compared the length of stay of IA and OA for admission with DRG 427. The mean IA length of stay (7.00 ± 4.32 days) was longer than the mean OA length of stay (6.10 ± 4.15 days). The difference, statistically significant, was 0.91 days (95% CI 0.57–1.25; *p* < 0.001). Moreover, IA hospitalization was longer than OA hospitalization both for males (6.80 days versus 6.07 days) and females (7.27 days versus 6.12 days).

Considering the length of stay, we analyzed the age breakdown for patients with DRG 427. The results are presented in Figure 4. We can observe that the mean length of stay is longer for IA than OA for all age groups except for the group <36 years.

### 3.3. Hypertension

We conducted a longitudinal comparison between IA and RA of the same patients with DRG 401, through a paired *t*-test. Excluding the outliers above 20 days, mean RA hospitalization (7.32 ± 6.77 days) was longer than mean IA hospitalization (5.24 ± 4.19 days). The difference, statistically significant, was 2.08 days (95% CI 1.58–2.57; *p* < 0.001).

Using a *t*-test for independent samples, we compared the length of stay of IA and OA for admission with DRG 401. The mean IA length of stay (5.99 ± 4.43 days) was longer than the mean OA length of stay (5.71 ± 4.20 days). The difference, statistically significant, was of 0.28 days (95% CI −0.01–1.25; *p* < 0.05). Moreover, IA hospitalization was longer than OA hospitalization for females (6.67 days IA versus 5.73 days OA), and for males OA was slightly longer than IA (5.58 days IA versus 5.70 days OA).

Concerning the length of stay, we analyzed the age breakdown for patients with DRG 401. The results are presented in Figure 5. We can observe that the mean length of stay is longer for IA than OA for all age groups except for the group <36 years.

### 3.4. Multivariate Logistic Regression Results

Multivariate logistic regression data highlight that all variables are statistically significant (*p* ≤ 0.001) and represent a risk factor for readmissions (Odds Ratio >1). Considering the type of pathology, a positive association between COPD and readmissions (*p* < 0.001; Odds Ratio 1.20) was detected. The same result was not obtained for cardiac arrythmias where a not statistically significant negative association was shown (*p* = 0.05; Odds Ratio 0.89). Moreover, considering the hypertension disease, a statistically significant negative association was detected (*p* = 0.01, Odds Ratio 0.77).

The variables that most influence the readmissions are the presence of COPD pathologies; the 2018–2019 two-year period; the male sex; the presence of a medical DRG; and the age range between 55 and 77 years.

These data were supported by multicollinearity evaluation, where a mean VIF of 1.75 was detected. Considering VIF values ranging from 1.07 to 3.56, low collinearities were always observed.

All data are shown in Table 3.

## 4. Discussion

Results show a decrease in repeated admissions from 5.36% in 2018 to 4.46% in 2021. The reduction was particularly evident in 2020 (4.57%) and 2021 (4.46%), probably due to the reduced health care access during the COVID-19 pandemic. However, there was also a slight decrease before the pandemic, from 2018 (5.36%) to 2019 (5.33%).

Readmitted patients were more frequently male, older, and admitted with a medical DRG. Otherwise, single admitted patients were more frequently female, younger, and admitted with a surgical DRG.

Concerning the pathology groups, a positive association between COPD and readmission was detected. This phenomenon was not shown for cardiovascular diseases.

Hypertension and arrhythmia are common conditions that can be effectively managed at home with adequate compliance and adherence to treatment, maintaining a proactive attitude to prevent complications. By constantly monitoring blood pressure and heart rate, for example, individuals with hypertension and arrhythmia can live almost serenely with the condition and autonomously prevent exacerbations, maintaining optimal health without having to frequently visit healthcare facilities. This goal can be achieved by modifying lifestyle, including regular exercise, a healthy diet, stress reduction, and of course taking prescribed medications as directed. The causes leading patients with hypertension/arrhythmia to hospitalization are generally secondary causes due to poor management of the underlying condition (e.g., heart attack, coronary artery disease, organ damage, etc.) and to a lesser extent due to the main pathology.

Even in the case of COPD, patients can take measures to reduce the risk of exacerbations, such as quitting smoking, avoiding air pollution, and keeping up to date with vaccinations against respiratory viruses. With adequate education and self-care, patients with COPD can minimize the severity and frequency of exacerbations and improve overall quality of life. However, COPD presents a greater challenge in terms of territorial management of the patient. Although there are various therapeutic options for COPD, it can still lead to sudden exacerbations that can cause severe breathing difficulties and can even be life-threatening, requiring immediate treatment in a hospital setting to carefully monitor the patient, sometimes repeatedly.

We found that RA mean hospital stay was longer than IA ones (difference of 1.57 days, 95% CI 1.36–1.78 days, *p* < 0.001), and IA mean hospital stay was longer than OA ones (difference of 0,62 days, 95% CI 0.52–0.72 days, *p* < 0.001). A readmitted patient stays, on average, almost two and a half times as long compared to a single admitted patient, considering both index admission and repeated admission. This represents a heavy hospital resource utilization, about 10.200 hospital stay days more than single admissions, considering only 2021, which is the equivalent of a department of 30 beds with 95% bed occupancy. Tuscany Region classifies the optimal use of an ordinary bed in a ward as being occupied 95% of the time (equal to 346 hospital days/year). Data are obtained dividing the 10,200 days of hospitalization by the 346 days developed for a single bed.

It should be noted that we excluded outliers from the analysis because they were not representative of the overall sample, but they determined an additional bed occupation.

The analysis of the selected clinical conditions demonstrates similar results regarding IA and RA length of stay. For patients with COPD (DRG 4912x, 492xx, 4932x), the difference was not statistically significant, whereas the difference was statistically significant for arrhythmias (DRG 427) and hypertension (DRG 401). The fact that RA mean length of stay was longer than IA mean length of stay may suggest that patients’ conditions increase in complexity from the IA to the RA.

Readmissions are an important issue in healthcare. It is possible to consult 30-day readmission rates by connecting to the site of Management e Sanità (MeS) by Sant’Anna School of Pisa [2]. In 2021, Tuscany Region had the highest readmission rate among the adhering regions.

The reduction in readmission in 2020 and 2021 is visible in all the regions. The cause of this reduction may be the reduced health care access during the COVID-19 pandemic; therefore, it will be interesting to see the readmission rates in the coming years.

Focusing on Tuscany, in 2021 the AOUP readmission rate was slightly above Tuscany’s mean value.

The trend in Tuscany shows a decrease in readmissions for all the healthcare facilities in the last three years. Before the pandemic, AOUP was the facility with the highest number of readmissions (5.92%).

Analyzing Tuscany’s healthcare facilities overall, we can see that readmissions were stable before the COVID-19 pandemic.

Reducing avoidable readmissions is one of the strongly pursued goals of health policies, since it would improve quality and reduce health care expenditures. However, a direct link between readmissions and quality of healthcare has not been clearly demonstrated, despite numerous studies conducted for several decades already. However, the knowledge of readmissions constitutes important information in the perspective of healthcare planning, so the monitoring is a useful tool that can allow one to find hidden dynamics in the mechanisms of admission and discharge of patients from hospital facilities.

In this study, readmissions were considered in their generality, but these may show only gross changes in the hospital’s inpatient activity. Next, we analyzed which principal diagnoses are most associated with readmissions. Within the same category of principal diagnoses, we selected the three most representative diagnoses and investigated the characteristics that differentiate patients who encounter readmission from patients with the same diagnosis who do not encounter readmission. These features are interesting for two reasons. In fact, they allow us to characterize the type of patients who are admitted to the AOUP, improving healthcare planning. On the other hand, we could hypothesize targeting interventions toward these patients to reduce the incidence of repeat hospitalizations.

Regarding interventions to reduce readmissions, the literature shows that there are significant effects when several interventions are combined simultaneously, which should be part of a prevention strategy, especially for those categories of patients most at risk. However, it must be pointed out that most of these studies are conducted in the United States, so there is no certainty about their validity in our country as well, which has a radically different health care system.

In this study, we may highlight some advantages and disadvantages.

Thanks to our data, the AOUP may apply strategies, based on the activation of post-discharge emergency medicine clinics also from Emergency Department access (verification of therapeutic adherence). This may limit the use of new access in the presence of non-acute symptoms.

On the other hand, the organizational structure of the regional healthcare system clearly separates hospital assistance from the territorial sort. In this way, hospitals cannot regulate personal services (home assistance, territorial general medicine) in and out of the hospital.

## 5. Conclusions

The challenge in recent years has been to strengthen the continuity of care between hospitals and communities. The aging of the population and the increase in chronic conditions and life expectancy are important changes that the National Health System has to deal with. Health services have been organized over the years with “hospital-centered” logic, which is very efficient in providing timely and quality treatment in acute conditions, but often inadequate to ensure care to patients with many chronicities, who are often elderly and frail.

In January 2022, after the approval of the National Recovery and Resilience Plan (Piano Nazionale di Ripresa e Resilienza, PNRR), the Italian government allocated EUR 8.42 billion to regions and autonomous provinces for the implementation of interventions in the health sector (strengthening of proximity networks, facilities, telemedicine for territorial health care and digitalization of the National Health Service).

In this way, in the coming years, we will witness the strengthening of territorial facilities and the transition to a new model of healthcare, made up of processes and pathways in which the hospital is an integral part of the system. Bearing this in mind, quality indicators, such as readmissions, can be used in monitoring the performance of new models of care.

## Figures and Tables

**Figure 1 healthcare-11-00651-f001:**
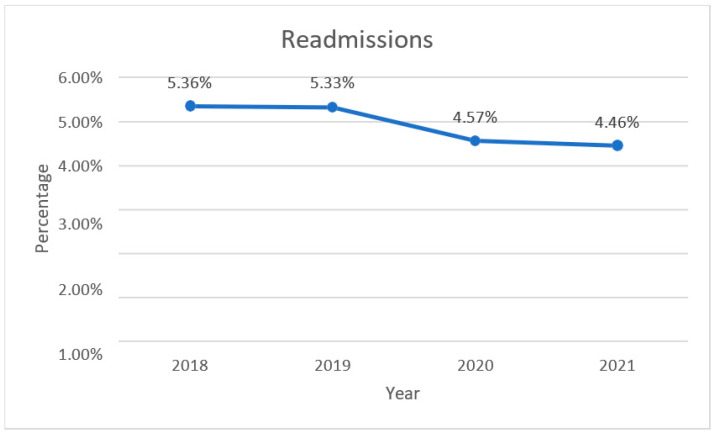
Trend of readmissions in AOUP 2018–2021.

**Figure 2 healthcare-11-00651-f002:**
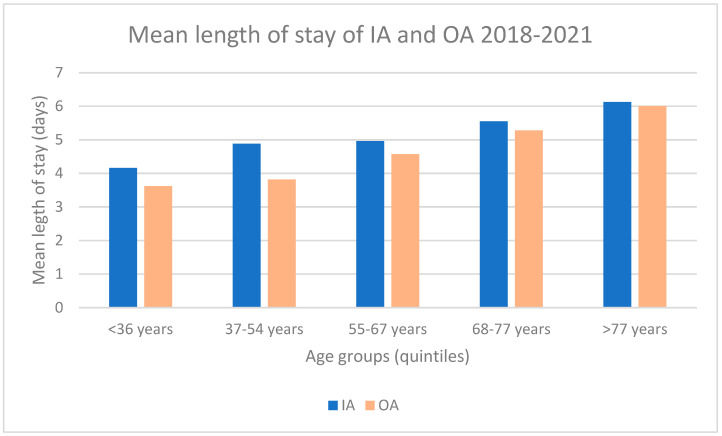
Comparison of the mean length of stay of IA and OA (2018–2021).

**Figure 3 healthcare-11-00651-f003:**
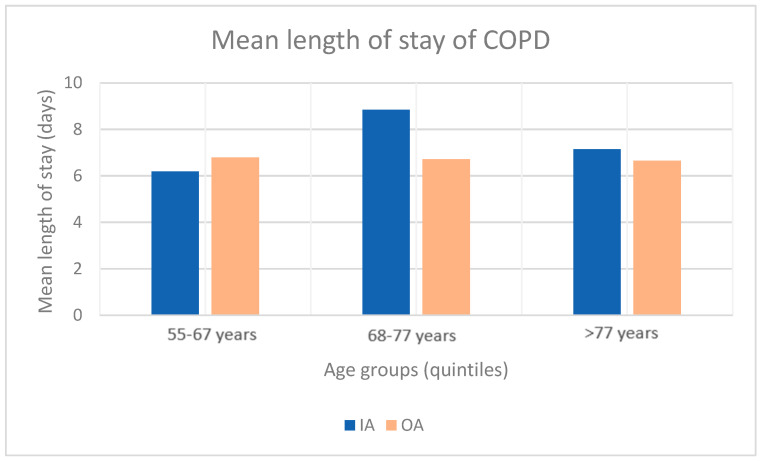
Comparison of the mean length of stay of COPD, IA and OA (2018–2021).

**Figure 4 healthcare-11-00651-f004:**
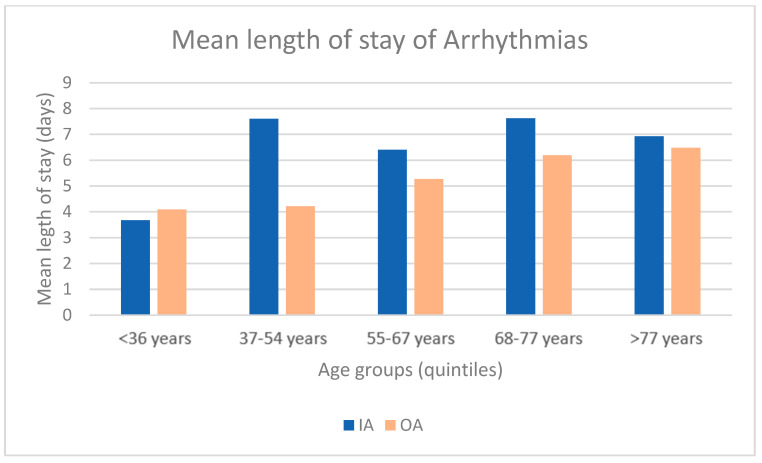
Comparison of the mean length of stay of Arrhythmias and OA (2018–2021).

**Figure 5 healthcare-11-00651-f005:**
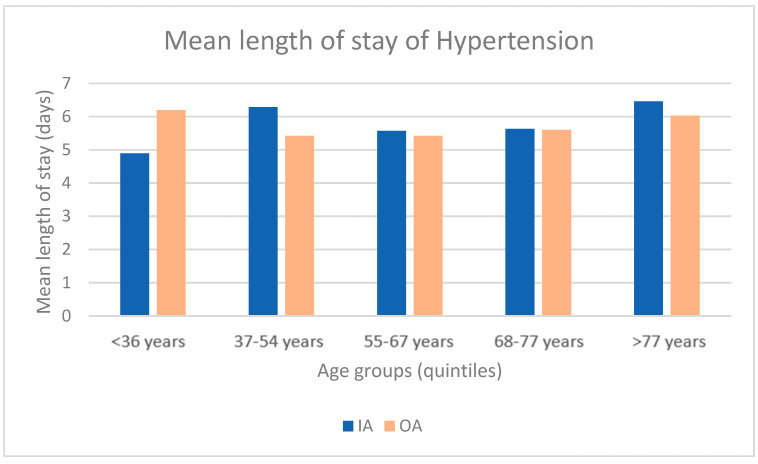
Comparison of the mean length of stay of hypertension and OA (2018–2021).

**Table 1 healthcare-11-00651-t001:** Clinical and demographic risk factors of readmissions.

Clinical Risk Factors	Demographic Risk Factors
Use of high-risk medications	Age
Polypharmacy	Prior hospitalization within the last 6 to 12 months
Number, duration and control of chronic conditions	Low health literacy
Specific clinical conditions (e.g., COPD, diabetes, heart failure, stroke, weight loss, depression, sepsis)	Low socio-economic status
	Reduced social network
	Distance from healthcare facilities
	Inefficient caregiver training

**Table 2 healthcare-11-00651-t002:** Comparison between IA, RA and OA according to the analyzed variables.

Variables	IA ^a^ N (%)	RA ^b^ N (%)	OA ^c^ N (%)	*p*-Value
Number	6727 (4.01)	8346 (4.97)	152,719 (91.02)	
Year				
2018	2067 (4.38)	2531 (5.36)	42,616 (90.26)	<0.001
2019	1954 (4.24)	2455 (5.33)	41,687 (90.44)	
2020	1297 (3.67)	1613 (4.57)	32,422 (91.76)	
2021	1409 (3.60)	1747 (4.46)	35,994 (91.94)	
Sex				
male	3601 (53.53)	4552 (54.54)	71,351 (46.72)	<0.001
female	3126 (46.47)	3794 (45.46)	81,368 (53.28)	
Age				
<36 years	895 (13.30)	1355 (16.24)	29,578 (19.37)	<0.001
37–54 years	1083 (16.10)	1331 (15.95)	29,891 (19.57)	
55–67 years	1441 (21.24)	1817 (21.77)	30,406 (19.91)	
68–77 years	1454 (21.61)	1740 (20.85)	29,279 (19.17)	
>77 years	1854 (27.56)	2103 (25.20)	33,564 (21.98)	
Type of DRG				
medical	5016 (74.57)	5441 (65.19)	67,811 (44.40)	<0.001
surgical	1711 (25.43)	2905 (34.81)	84,908 (55.60)	
Clinical diagnosis				
COPD ^d^	393 (6.67)	483 (8.20)	5012 (85.12)	<0.001
Cardiac arrhythmias ^e^	306 (4.1)	335 (4.49)	6818 (91.41)	n.s.
Hypertension ^f^	899 (3.94)	944 (4.14)	20,949 (91.91)	<0.001
	IA Mean (SD)	RA Mean (SD)	OA Mean (SD)	*p*-value
Length of stay (days) ^g^	5.3 (4.41)	5.36 (4.27)	4.68 (4.01)	<0.01
Age	61.7 (20.9)	59.4 (22.4)	55.9 (24.3)	<0.001

^a^ IA (index admissions) indicates the first admission of patients that were readmitted with the same MDC within 30 days after discharge. ^b^ RA (repeated admissions) indicates admissions that followed the first one within 30 days from the previous discharge with the same MDC. ^c^ OA (only admissions) indicates a single admission that is not followed by another within 30 days from discharge with the same MDC. ^d^ COPD was defined as discharge with DRG code 4912x, 492xx, 4932x. ^e^ Cardiac arrhythmias were defined as discharge with DRG code 427. ^f^ Hypertension was defined as discharge with DRG code 401. ^g^ Outliers above the 95th percentile were excluded from the analysis regarding the length of stay.

**Table 3 healthcare-11-00651-t003:** Multivariate logistic regression data.

Variables	Odds Ratio	*p*-Value	95% [CI]	VIF	1/VIF
Years Group 2018–2019 2020–2021	1.2 1	0.001	[1.11–1.29]		
Sex male female	1.26 1	0.001	[1.20–1.33]	1.85	0.54
Age <36 years 37–54 years 55–67 years 68–77 years >77 years	1 1.71 2.23 2.20 1.97	0.001 0.001 0.001 0.001	[1.56–1.87] [2.04–2.43] [2.01–2.40] [1.80–2.15]	1 1.5 1.58 1.68 0.23	1 0.67 0.63 0.59 0.49
Type of DRG medical surgical	3.95 1	<0.001	[3.74–4.19]	3.56	0.28
COPD yes no	1.20 1	<0.001	[1.08–1.34]	1.07	0.93
Cardiac arrhythmias yes no	0.89 1	0.05	[0.81–0.97]	1.19	0.84
Hypertension yes no	0.77 1	0.01	[0.72–0.83]	1.26	0.79

## Data Availability

The datasets used and/or analyzed during the current study are available from the corresponding author on reasonable request.

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
