# Peer review of "Evaluation, Description and Magnitude of Readmission Phenomenon in Azienda Ospedaliero Universitaria Pisana (AOUP) for Chronic-Degenerative Diseases in the Period 2018–2021"

_healthcare, 2023, doi:10.3390/healthcare11050651_

Round 1

Reviewer 1 Report

The manuscript is well written and the topic is of interest. However, the main weakness of the study is the study period, it includes Covid-19 pandemic, so the results may be distorted and we do not know up to what extent. It would be quite necessary to compare two periods, one including Covid-19 pandemic and the other one excluding it. 

According to the paragraph included in the Introduction section 'In 2000, SDO was updated with ICD-9-CM classification, [20] and it contains patient’s personal data, hospitalization features and clinical information The importance of SDO is about a uniform classification of hospital discharges, feeding a huge informative flow that allows economic, management, clinic, epidemiologic evaluations.', tt seems like there is data available from 2000. Why didn't you take data from earlier year instead of 2018? If you have done so, it would had been possible to see the % change in readmissions in the years previous to Covid-19 pandemic.

Materials and Methods:

- Major Diagnostic Category (MDC). This is already defined.

- You mention that the outliers were excluded from the analysis. However, some descriptive statistics on outliers are missing, such as how many outliers were found, age, disease condition, gender, etc.

Results:

- "74.57% surgical DRG versus 25.43% medical DRG". Actually, it is 74.57% medical DRG versus 25.43% surgical DRG.

- Were the differences between age groups in the mean leangth of stay of IA and OA statistically significant?

Discussion:

- Please, revise the percentages associated to the years. They are few mistakes. "The results show a decrease in repeated admissions from 5.36% in 2021 to 4.46% in 2021. The reduction was particularly evident in 2020 (4.57%) and 2020 (4.46%), probably due to the reduced health care access during the Covid-19 pandemic. However, there was also a slight decrease before the pandemic, from 2018 (5.33%) to 2019 (5.36%)."

- "This represents a heavy hospital resource utilization, about 10.200 hospital stay days more than single admissions, considering only 2021, which is the equivalent of a department of 30 beds with 95% bed occupancy." Please, explain how was this estimated.

- Please, give som thoughts on advantages and disadvantages of the study.

Conclusions:

- The conclusions section is very large. Please, comment very briefly the conclusion of the study and move all the other staff into results. 

Author Response

Dear Reviewer,

Thank you for your comments.

The manuscript is well written and the topic is of interest. However, the main weakness of the study is the study period, it includes Covid-19 pandemic, so the results may be distorted and we do not know up to what extent. It would be quite necessary to compare two periods, one including Covid-19 pandemic and the other one excluding it. 

According to the paragraph included in the Introduction section 'In 2000, SDO was updated with ICD-9-CM classification, [20] and it contains patient’s personal data, hospitalization features and clinical information The importance of SDO is about a uniform classification of hospital discharges, feeding a huge informative flow that allows economic, management, clinic, epidemiologic evaluations.', tt seems like there is data available from 2000. Why didn't you take data from earlier year instead of 2018? If you have done so, it would had been possible to see the % change in readmissions in the years previous to Covid-19 pandemic.

Patients discharged in 2018-2019 and 2020-2021 were considered for the following reasons:

- to have a representative cohort of patients discharged in the available pre-pandemic two-year period and in the pandemic two-year period;

- the years choice was influenced by organizational reasons. In fact, there was substantial stability in our national ministerial criteria for appropriateness of the setting care used by the healthcare units (ordinary hospitalization rather than outpatient procedure);

- In Tuscany during the year 2018 the SDO coding guidelines were changed (DGRT 773/2018). Therefore the attribution of a patient to a MDC could have been different with the application of previous versions of guidelines

Materials and Methods:

- Major Diagnostic Category (MDC). This is already defined.

Correction has been applied

- You mention that the outliers were excluded from the analysis. However, some descriptive statistics on outliers are missing, such as how many outliers were found, age, disease condition, gender, etc.

Outliers were excluded following a temporal criterion (above 19 days). Outliers are verified separately as they may not be significant. Usually they are excluded from the 95th percentile (through a frequency distribution). In the text we added some information about the size of this cohort (5.6%).

Results:

- "74.57% surgical DRG versus 25.43% medical DRG". Actually, it is 74.57% medical DRG versus 25.43% surgical DRG.

Correction has been done

- Were the differences between age groups in the mean leangth of stay of IA and OA statistically significant?

 Statistical data have been added

Discussion:

- Please, revise the percentages associated to the years. They are few mistakes. "The results show a decrease in repeated admissions from 5.36% in 2021 to 4.46% in 2021. The reduction was particularly evident in 2020 (4.57%) and 2020 (4.46%), probably due to the reduced health care access during the Covid-19 pandemic. However, there was also a slight decrease before the pandemic, from 2018 (5.33%) to 2019 (5.36%)."

Sentences have been corrected

- "This represents a heavy hospital resource utilization, about 10.200 hospital stay days more than single admissions, considering only 2021, which is the equivalent of a department of 30 beds with 95% bed occupancy." Please, explain how was this estimated.

Tuscany Region defines the optimal use of an ordinary beds in ward when it is 95% occupied (equal to 346 hospital days/year). Data is obtained dividing the 10,200 days of hospitalization for 346 days developed for a single bed. A ward of about 30 beds is obtained.

It has been added in the text.

- Please, give some thoughts on advantages and disadvantages of the study.

Advantages and disadvantages have been added in Discussion section.

Conclusions:

- The conclusions section is very large. Please, comment very briefly the conclusion of the study and move all the other staff into results.

Conclusion section has been edited as requested. 

For more details please see the revised manuscript file. Thanks in advance.

Reviewer 2 Report

An interesting article on readmissions in the inpatient setting. The topic is clinically relevant, as cost-effectiveness is becoming increasingly important in medicine.

- Paragraph methods (example)

“30days” “USA)and”

Please change the spaces throughout the text. There are many errors there.

Corrections to the structure

- Paragraph 3.1 COPD (example; Please change in the complete text.)

“1.62 (95% CI 0.64 – 2.60) days, p < 0.001.”

Better is:

1.62 days (95% CI 0.64 – 2.60; p < 0.001).”

Paragraph results and conclusion

- Bulleted lists should not be used in body text. This should be elaborated and formulated in a continuous text.

Corrections to the content

- Please state the significance level under methods.

- The focus is on COPD, cardiac arrhythmias and hypertension. The title of the article should be adapted according to this focus.

Author Response

An interesting article on readmissions in the inpatient setting. The topic is clinically relevant, as cost-effectiveness is becoming increasingly important in medicine.

- Paragraph methods (example)

“30days” “USA)and”

Please change the spaces throughout the text. There are many errors there.

 Corrections have been made

Corrections to the structure

- Paragraph 3.1 COPD (example; Please change in the complete text.)

Correction has been made

“1.62 (95% CI 0.64 – 2.60) days, p < 0.001.”

Better is:

1.62 days (95% CI 0.64 – 2.60; p < 0.001).”

 Corrections have been made

Paragraph results and conclusion

- Bulleted lists should not be used in body text. This should be elaborated and formulated in a continuous text.

 Conclusion section has been modified and bulleted list has been removed

Corrections to the content

- Please state the significance level under methods.

Significance level of chi squadre test has been added in methods section

- The focus is on COPD, cardiac arrhythmias and hypertension. The title of the article should be adapted according to this focus.

Title has been changed in “Evaluation, description and magnitude of readmission phenomenon in Azienda Ospedaliero- Universitaria Pisana (AOUP) for chronic-degenerative diseases in the period 2018-2021”

Reviewer 3 Report

There are some grammatical errors in the abstract "the length ofstay of index hospitaliza-" and "he territory,which" needs a space between "," and "which"

Introduction:

"In last decades," should be "In the last decade"

"In the U.S.,", I believe United States should be spelled out first and then acronyms can be used later. 

"payment ofhealthcare" there needs to be a space between 'of' and 'healthcare'

You had a new paragraph started, but I believe it should be part of the previous paragraph. 

I like the table of risk factors of readmissions. 

Methods:

"within 30days from" there needs to be a spaced between '30' and 'days'

I would like to see some information about the consent of the participants or how data was obtained from the hospital i.e. were patients notified, was there an agreement with the hospital, did IRB approve the study? 

Results:

Very detailed and easy to follow for our readers. In your table, you are missing '.' and have ',' . If the numbers are supposed to be for SD please move to that part of the table instead of the table with N)%) instead. Please fix. Also, why was your SD so for from the mean? There is a big difference from a possible 31.6 to 80.2. 

Your charts were easy to read and very informative. 

Author Response

Dear Reviewer,

There are some grammatical errors in the abstract "the length ofstay of index hospitaliza-" and "he territory,which" needs a space between "," and "which"

Sentences have been corrected

Introduction:

"In last decades," should be "In the last decade"

It has been corrected

"In the U.S.,", I believe United States should be spelled out first and then acronyms can be used later. 

It has been changed

"payment ofhealthcare" there needs to be a space between 'of' and 'healthcare'

You had a new paragraph started, but I believe it should be part of the previous paragraph. 

It has been corrected

Methods:

"within 30days from" there needs to be a spaced between '30' and 'days'

It has been corrected

I would like to see some information about the consent of the participants or how data was obtained from the hospital i.e. were patients notified, was there an agreement with the hospital, did IRB approve the study? 

Data were obtained by the administrative HDR database of the AOUP. All records were coded with a medical DRG. Considering the use of anonymous data we deemed sufficient the application of an informed consent for patients and healthcare personnel. This information has been added in the text.

Results:

Very detailed and easy to follow for our readers. In your table, you are missing '.' and have ',' . If the numbers are supposed to be for SD please move to that part of the table instead of the table with instead. Please fix. Also, why was your SD so for from the mean? There is a big difference from a possible 31.6 to 80.2. 

Table has been edited as requested. Differences between mean and SD is due to the sample variability. 

Round 2

Reviewer 1 Report

I am not convinced about the relevance of the study as it includes the covid-19 pandemic period.

Author Response

Dear Reviewer,

we recognize that COVID period influenced the hospitalization and readmission processes. This may be evaluated by variance analysis. For this reason we added one-way ANOVA test in order to analyze some difference in readmission phenomenon between the years 2018-19 and 2020-21.

We highlight a statistically significat decrease of readmissions in COVID pandemic period in comparion to pre-COVID years (p= 0.01).

All these considerations have been included in the text.
